# Cross-species evaluation of TANGO2 homologs, including HRG-9 and HRG-10 in *Caenorhabditis elegans,* challenges a proposed role in heme trafficking

Sarah E Sandkuhler[1], Kayla S Youngs[2], Olivia Gottipalli[3], Laura D Owlett[2], Monica B Bandora[4], Aaliya Naaz[5], Euri Kim[6], Lili Wang[7], Andrew Wojtovich[8], Vandana Gupta[6], Michael Sacher[5,9], Samuel J Mackenzie[2]*

[1]Department of Pathology, University of Rochester Medical Center, Rochester, United States; [2]Department of Neurology, University of Rochester Medical Center, Rochester, United States; [3]Emory University, Atlanta, United States; [4]Morgan State University, Baltimore, United States; [5]Department of Biology, Concordia University, Montreal, Canada; [6]Department of Medicine, Brigham and Women's Hospital Harvard Medical School, Boston, United States; [7]Department of Pharmacology, Vanderbilt University, Nashville, United States; [8]Department of Anesthesiology and Perioperative Medicine, University of Rochester Medical Center, Rochester, United States; [9]Department of Anatomy and Cell Biology, McGill University, Montreal, Canada

**\*For correspondence:**
samuel_mackenzie@urmc.
rochester.edu

## eLife Assessment

This **valuable** study provides **solid** evidence that supports TANGO2 homologs, including HRG-9 and HRG-10, can play a role in cellular bioenergetics and oxidative stress homeostasis. It also challenges the previously reported role of TANGO2 in heme transport and paves the way for future mechanistic studies addressing the mechanisms of how TANGO2 regulates oxidative stress homeostasis. The strengths include the use of different model systems, genetic tools, behavioral assays, and efforts by the authors in using the same reagents to reproduce results of other groups.

**Abstract** Mutations in the *TANGO2* gene are associated with a severe neurometabolic disorder in humans, often presenting with life-threatening metabolic crisis. However, the function of TANGO2 protein remains unknown. It has recently been proposed that TANGO2 transports heme within and between cells, from areas with high heme concentrations to those with lower concentrations. Here, we demonstrate that prior heme-related observations in *Caenorhabditis elegans* lacking TANGO2 homologs HRG-9 and HRG-10 may be better explained by a previously unreported metabolic phenotype, characterized by reduced feeding, decreased lifespan and brood sizes, and poor motility. We also show that several genes not implicated in heme transport are upregulated in the low heme state and conversely demonstrate that *hrg-9* in particular is highly responsive to oxidative stress, independent of heme status. Collectively, these data implicate bioenergetic failure and oxidative stress as potential factors in the pathophysiology of TANGO2 deficiency, in alignment with observations from human patients. Our group performed several experiments in yeast and zebrafish deficient in TANGO2 homologs and was unable to replicate prior findings from these models. Overall, we believe there is insufficient evidence to support heme transport as the primary function for TANGO2.

## Introduction

Transport and Golgi Organization 2 (TANGO2) Deficiency Disorder (TDD) is a severe, progressive, neurodegenerative disease primarily affecting children and young adults, frequently accompanied by recurrent, life-threatening metabolic crises (*Kremer et al., 2016*; *Lalani et al., 2016*; *Powell et al., 2021*; *Jennions et al., 2019*). While TANGO2 is conserved in species across all three domains of life, its cellular function has not been fully elucidated. Initially identified in a screen of *Drosophila* genes, which, when depleted, precipitate ER-Golgi fusion defects (*Bard et al., 2006*), TANGO2 has since been shown to localize to both the cytoplasm and the mitochondria (*Milev et al., 2021*; *Lujan et al., 2025*). Mutations in TANGO2 have been associated with altered mitochondrial respiration and morphology (*Heiman et al., 2022*), disregulated lipid homeostasis (*Lujan et al., 2023*; *Mehranfar et al., 2024*), and defects in fatty acid oxidation (*Heiman et al., 2022*) in multiple models. Treatment with supplemental pantothenic acid, the precursor of coenzyme A (CoA), has been shown to improve lipid profile abnormalities in cells (*Mehranfar et al., 2024*), rescue phenotypic defects in a *Drosophila* TDD model (*Asadi et al., 2023*), and potentially reduce metabolic crises in patients (*Miyake et al., 2023*), implicating reduced CoA, lipid dysregulation, and impaired mitochondrial function as putative drivers of TDD symptoms.

A recent study by Sun et al. suggested that two *Caenorhabditis elegans* homologs of TANGO2, HRG-9 and HRG-10 (**H**eme-**R**esponsive **G**ene), play a critical role in heme export from intestinal cells and lysosome-related organelles and that TANGO2 exhibits a similar function in mobilizing heme from mitochondria in other eukaryotic organisms including yeast (*Sun et al., 2022*). Two subsequent studies also reported that TANGO2 is involved in heme transport, albeit via discordant mechanisms (*Han et al., 2023*; *Jayaram et al., 2025*). The notion that defective heme transport underlies the pathophysiology of TANGO2 deficiency marks a sizeable shift from the emerging scientific consensus that TANGO2 is critical for CoA and lipid homeostasis. Therefore, our primary objective was to validate key findings pertaining to heme transport in *C. elegans* while also replicating key findings in zebrafish and yeast lacking TANGO2 homologs. We hypothesized that, if present, defects in heme transport may be downstream features of aberrant cellular metabolism and thus may not be central to the pathophysiology of TDD.

## Results

### *C. elegans* lacking TANGO2 homologs (double knockout [DKO]) demonstrate modest survival benefit upon toxic heme analog exposure

We first sought to validate results from prior experiments using heme analogs in worms lacking both HRG-9 and HRG-10 (DKO). In previous studies, DKO worms exposed to 1 µM concentration of the toxic heme analog gallium protoporphyrin IX (GaPP) showed a significant survival benefit in contrast with nearly uniform lethality in wild type (Bristol N2) worms (*Sun et al., 2022*). We exposed gravid DKO and N2 nematodes to GaPP at varying concentrations, removing $P_0$ worms at 24 hours and counting alive and dead $F_1$ progeny at 72 hours. While we required relatively higher concentrations of GaPP to achieve lethality, all strains exhibited a clear dose-dependent reduction in survival (*Figure 1A*). A statistically significant group effect was observed at 2 µM, with DKO worms showing a modest relative survival benefit compared to N2.

### DKO *C. elegans* demonstrate increased lawn avoidance and reduced pharyngeal pumping

In maintaining the DKO strain under basal conditions (i.e., no GaPP exposure), we incidentally observed relatively intact *Escherichia coli* lawns on plates housing DKO worms compared to N2 plates. On closer examination, DKO worms demonstrated several previously undescribed phenotypic features including lawn avoidance, a greater propensity for crawling off plates, reduced pharyngeal pumping, and decreased survival (*Figure 2A, B and F*). Based on these observations, we hypothesized that the GaPP survival difference in our initial experiment might have been driven, at least in part, by reduced GaPP consumption. We therefore also examined the effect of GaPP exposure on worms lacking a pharyngeal acetylcholine receptor subunit (*eat-2*(*ad465*)). The *eat-2* strain exhibits reduced pharyngeal pumping and has been used extensively as a model of dietary restriction (*Lakowski and*

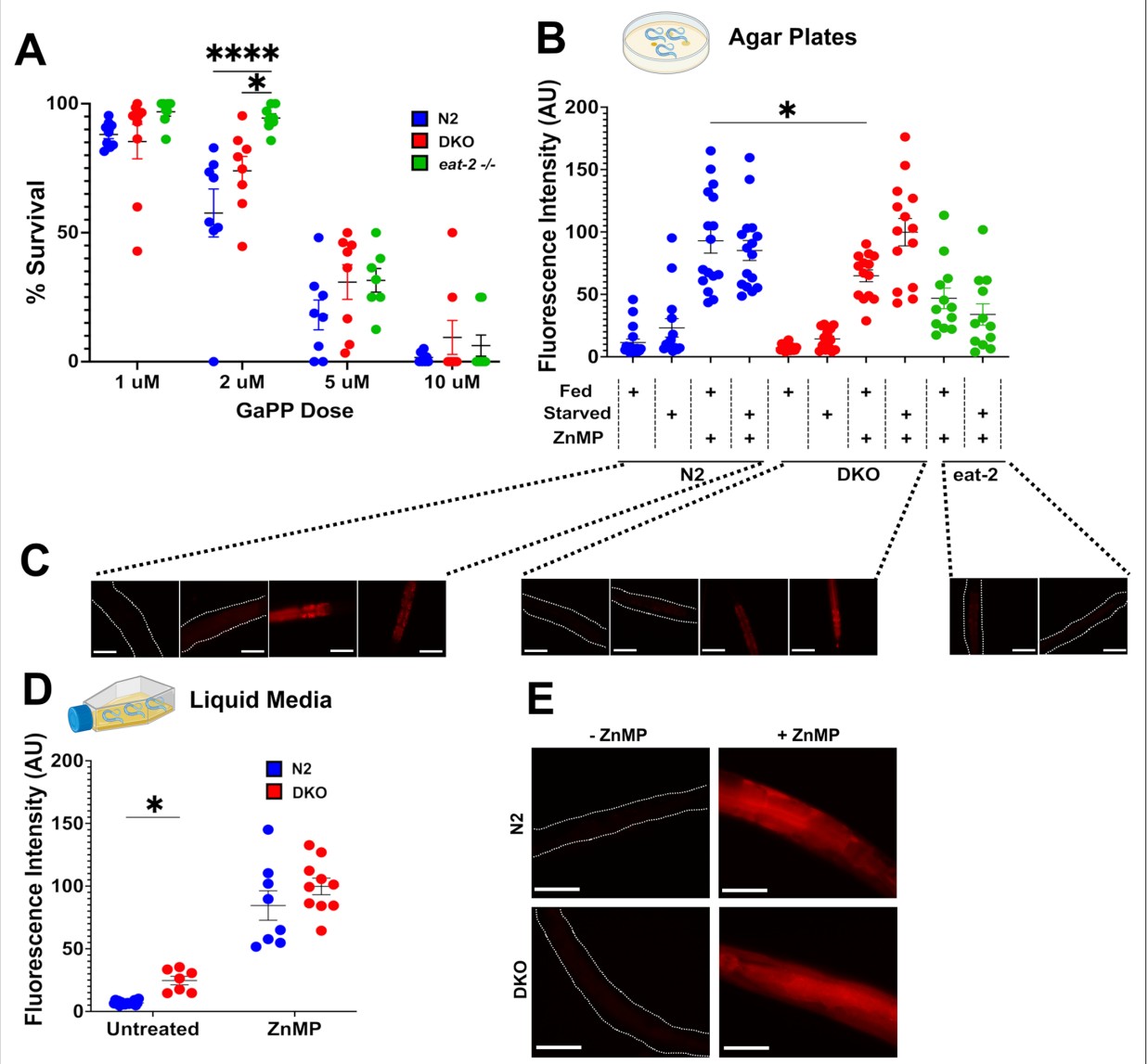

**Figure 1.** Reduced ingestion of toxic and fluorescent heme analogs may explain differences observed between wildtype *C. elegans* and worms lacking HRG-9 and HRG-10 (double knockout [DKO]). (**A**) Survival of N2, DKO, or *eat-2* knockout worms exposed to 1, 2, 5, or 10 µM GaPP for 72 hours. Each dot represents the number of offspring laid by one adult worm on one GaPP-treated plate after 24 hours. N=3 independent experiments. *p<0.05, ****p<0.0001. (**B**) Quantification of fluorescent staining in N2, DKO, and *eat-2* worms grown on OP50 *E. coli* plates under fed and starved conditions with or without 40 µM ZnMP treatment. A.U. arbitrary units, N=15 worms analyzed over three independent experiments. *p<0.05. (**C**) Representative images of red fluorescence in N2, DKO, and *eat-2* worms grown on OP50 *E. coli* plates. Scale bars represent 50 µM. (**D**) Quantification of fluorescent staining in N2 and DKO cultured in normal heme axenic media with or without 40 µM ZnMP treatment. A.U. arbitrary units, N=7–15 worms analyzed over three independent experiments. *p<0.05. (**E**) Representative images of red fluorescence in N2 and DKO worms grown in liquid media. Scale bars represent 50 µM.

*Hekimi, 1998*; *Greer and Brunet, 2009*). Of the three strains tested, survival was highest in *eat-2* mutants (*Figure 1A*) despite the fact that these worms have no known defects in heme metabolism or transport.

## DKO *C. elegans* demonstrate reduced ZnMP fluorescence

DKO nematodes at the L4 larval stage were previously shown to accumulate the fluorescent heme analog zinc mesoporphyrin IX (ZnMP) in intestinal cells in low-heme (4 µM) liquid media. While attempting to replicate this experiment, we observed that both wildtype and DKO nematodes entered L1 arrest under these conditions. Therefore, to allow for developmental progression, we grew

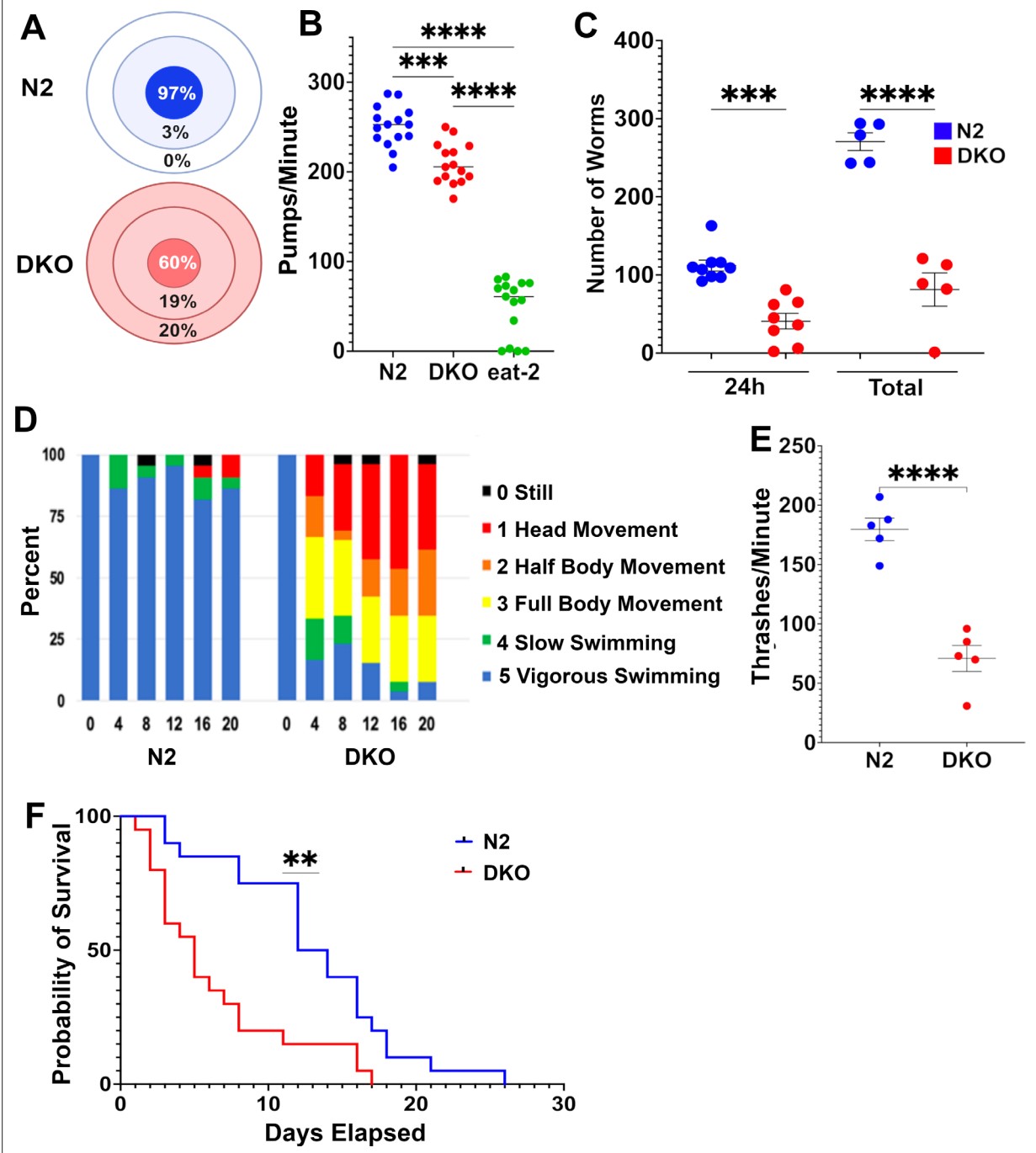

**Figure 2.** DKO nematodes demonstrate lawn avoidance and reduced pharyngeal pumping, brood sizes, motility, and survival. (**A**) Proportion of N2 or DKO worms present on OP50 lawn (innermost ring), off OP50 lawn (middle ring), or missing or dead from NGM plate (outer ring). N=180 worms over three independent experiments. (**B**) Number of pharyngeal pumps in a 1-minute period in N2, DKO, and *eat-2* knockout worms. N=15 worms over three independent experiments. (**C**) Number of viable offspring laid by single adult N2 or DKO worms either after 24 hours of egg lay or across total egg-laying period (5 days). N=8 broods for 24 hour counts, N=5 broods for total brood size counts. (**D**) Swimming behavior of N2 and DKO worms over a 20-minute interval. Worms were observed at 4-minute intervals and scored from 0 to 5 on swimming intensity. Bars represent the proportion of worms at each score. (**E**) Quantification of thrashes after 4 minutes in M9 buffer. (**F**) Longevity of N2 and DKO *C. elegans* observed from L4 larval stage. N=30 worms over three independent experiments (**p<0.01, ***p<0.001, ****p<0.0001).

worms on standard OP50 *E. coli* plates and in media containing physiological levels of heme (20 μM). We then examined whether differences in ZnMP uptake persisted under these basal conditions. DKO worms grown on ZnMP-treated *E. coli* plates displayed significantly reduced intestinal ZnMP fluorescence compared to N2 (*Figure 1B and C*). Using basal heme media with ZnMP, there was no significant difference in ZnMP fluorescence between DKO and wildtype nematodes, although DKO worms grown in media without ZnMP exhibited significantly higher autofluorescence (*Figure 1D and E*). To test whether autofluorescence may have contributed to the higher fluorescent intensities previously reported in heme-deficient DKO worms, we repeated this experiment on agar plates under starved conditions but did not observe a difference between groups (*Figure 1B*).

## DKO *C. elegans* demonstrate multiple features suggestive of bioenergetic dysfunction

In assessing brood survival in the GaPP assay, we also observed significantly smaller starting broods for DKO nematodes, a previously unreported finding that persisted in the absence of GaPP (*Figure 2C*). Oogenesis and egg-laying require a high energetic expenditure for gravid *C. elegans*, and reduced brood size is a known feature of several metabolically impaired nematode strains (*Van Raamsdonk et al., 2010*; *Byrne et al., 2019*). We also observed reduced movement on the plate from DKO worms and decided to further characterize the worms' capacity for movement by subjecting them to a swim exhaustion assay. Nematodes were placed in isotonic M9 buffer and scored on their swimming ability at 4-minute intervals. While *C. elegans* are typically able to swim continuously for up to 90 minutes in M9 media (*Ghosh and Emmons, 2008*), we observed that the DKO worms quickly became exhausted and could not maintain normal swimming shortly after being placed in media (*Figure 2D*). We further

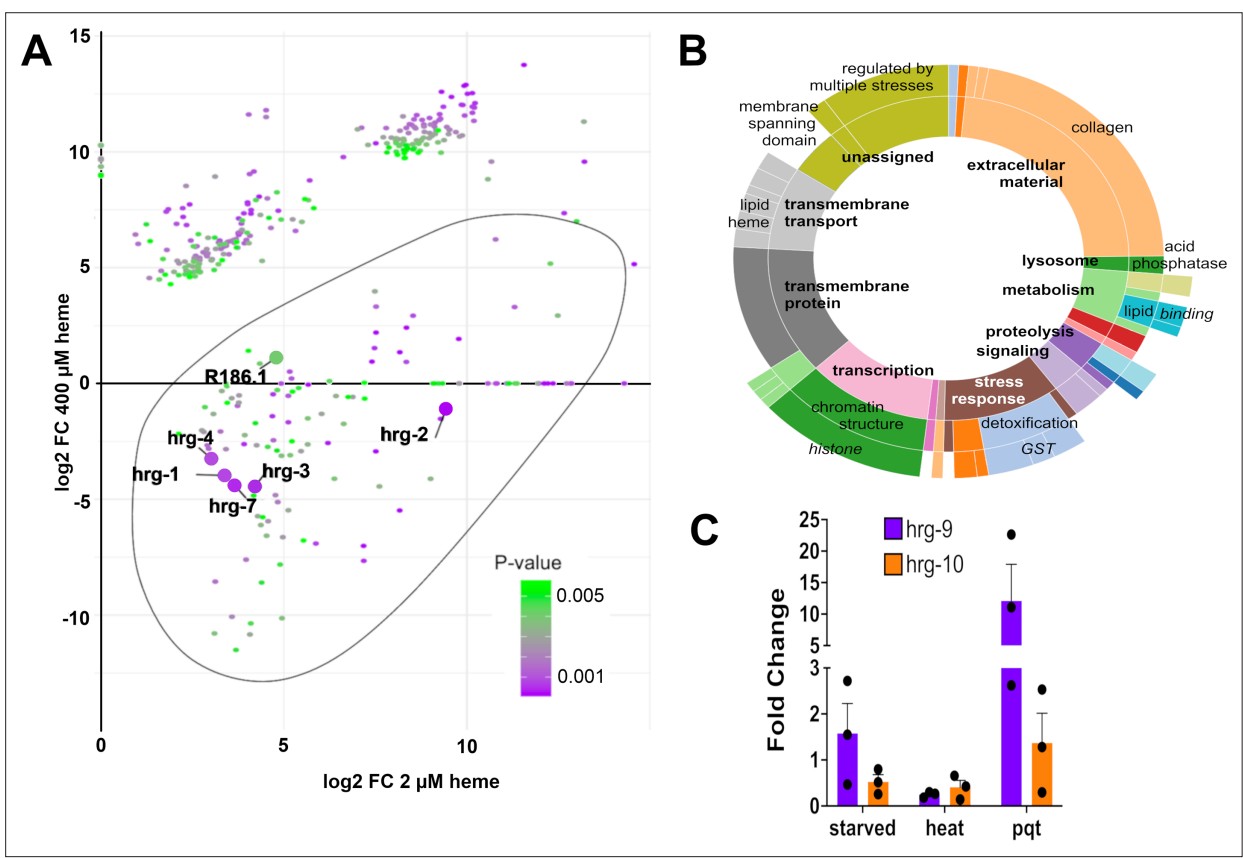

**Figure 3.** RNA-seq and qPCR analysis show that *hrg-9* and *hrg-10* are not uniquely heme responsive but instead may be preferentially induced under conditions of oxidative stress. (**A**) Analysis of top 500 genes with differential expression under low heme (2 μM) and high heme (400 μM) conditions. The outline represents 134 relevant genes identified by cluster analysis. R186.1 is the alternative sequence name for *hrg-9*. (**B**) Gene ontology analysis identified a variety of biological roles for genes within this cluster. (**C**) RT-qPCR of *hrg-9* and *hrg-10* under non-heme stress conditions: 24-hour starvation, 4-hour exposure to 34°C heat, and 25 mM paraquat (pqt). N=3 independent experiments.

quantified the rate of thrashing in M9 media during the first 4-minute interval and found that DKO worms thrashed 61% slower compared to N2 (*Figure 2E*).

## Oxidative stress is a driver of TANGO2 homolog transcription

Given previous observations implicating oxidative stress as a hallmark feature of TDD (*Heiman et al., 2022*), we next sought to examine what other genes were enriched under low heme conditions. We reanalyzed the RNA-seq dataset generated by Sun et al., employing the Empirical Analysis of Gene Expression in R (edgeR) package on raw counts to accurately perform between-group comparisons across low (2 μM), optimal (20 μM), and high (400 μM) heme conditions. We extracted the top 500 enriched genes and plotted those that showed significantly increased expression in the low heme state, based on computational clustering (N=134; *Figure 3A*). Several genes with no known heme-related functions demonstrated stronger relative expression and higher likelihood ratios of conditional effect than did *hrg-9*. Furthermore, gene ontology analysis of genes with similar enrichment to *hrg-9* revealed a wide spectrum of biological processes and cellular roles, including but not limited to collagen deposition, cellular detoxification, and lipid binding (*Figure 3B*). To test what alternate forms of cell stress might induce *hrg-9* and *hrg-10* expression, we exposed N2 nematodes to heat (34°C),

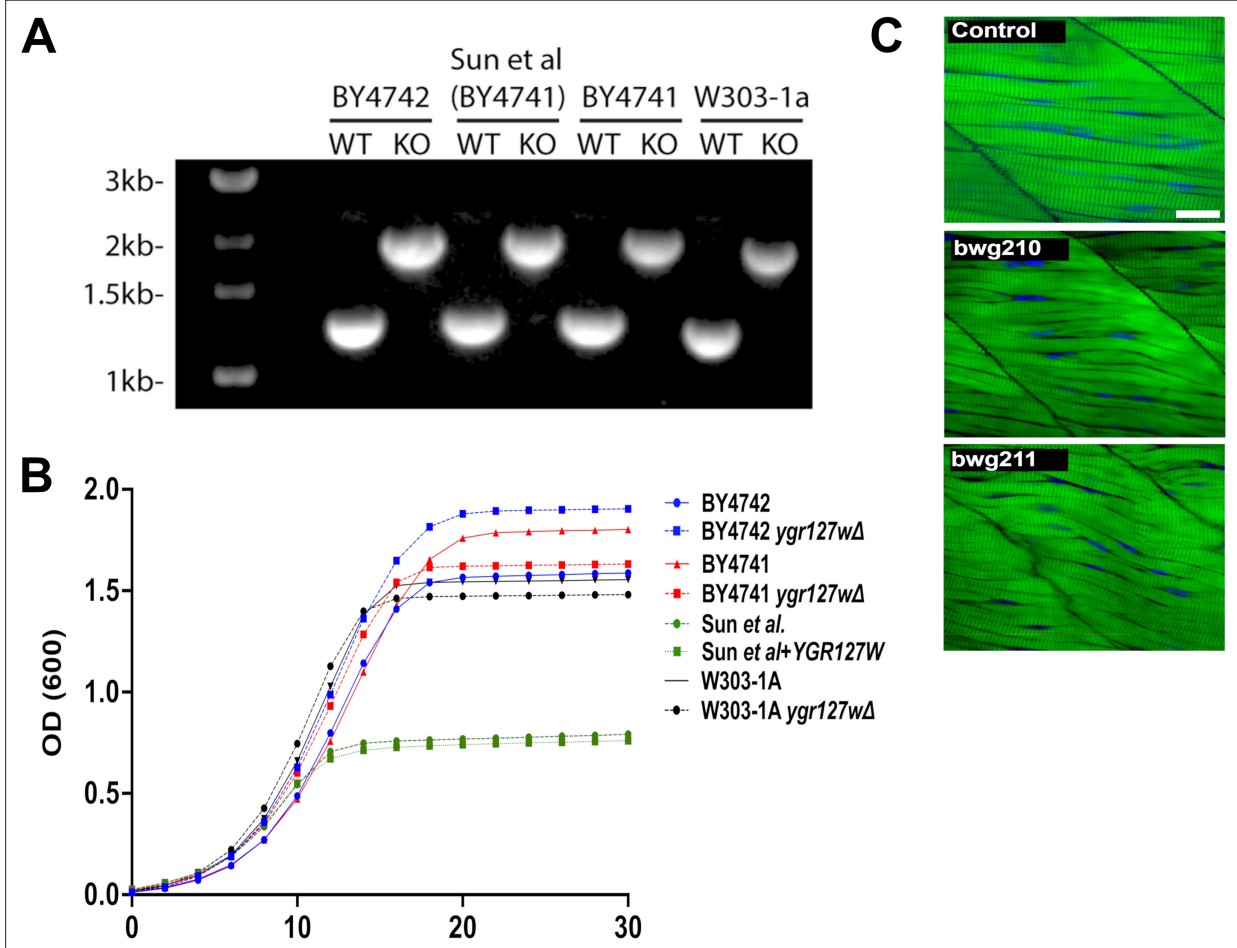

**Figure 4.** Growth and muscle fiber integrity in yeast and zebrafish models of TANGO2 deficiency fail to replicate previously reported phenotypes. (**A**) PCR confirmation of *YGR127w* knockout cassette integration across yeast strains. (**B**) Yeast growth curves. Different strains were grown in SC medium or SC medium lacking histidine at 25°C. (**B**) Whole-mount phalloidin staining of control and two strains of *tango2⁻/⁻* zebrafish (*bwg²¹⁰* and *bwg²¹¹*). Myofibers in mutants lack the parallel organization observed in controls but do not demonstrate significant myofiber breakdown or damage. Representative images; N=8–10 in each group. Scale bar = 5 mm.

The online version of this article includes the following source data for figure 4:

**Source data 1.** Original agarose gel in *Figure 4A*.

**Source data 2.** Original files for agarose gel in *Figure 4A*.

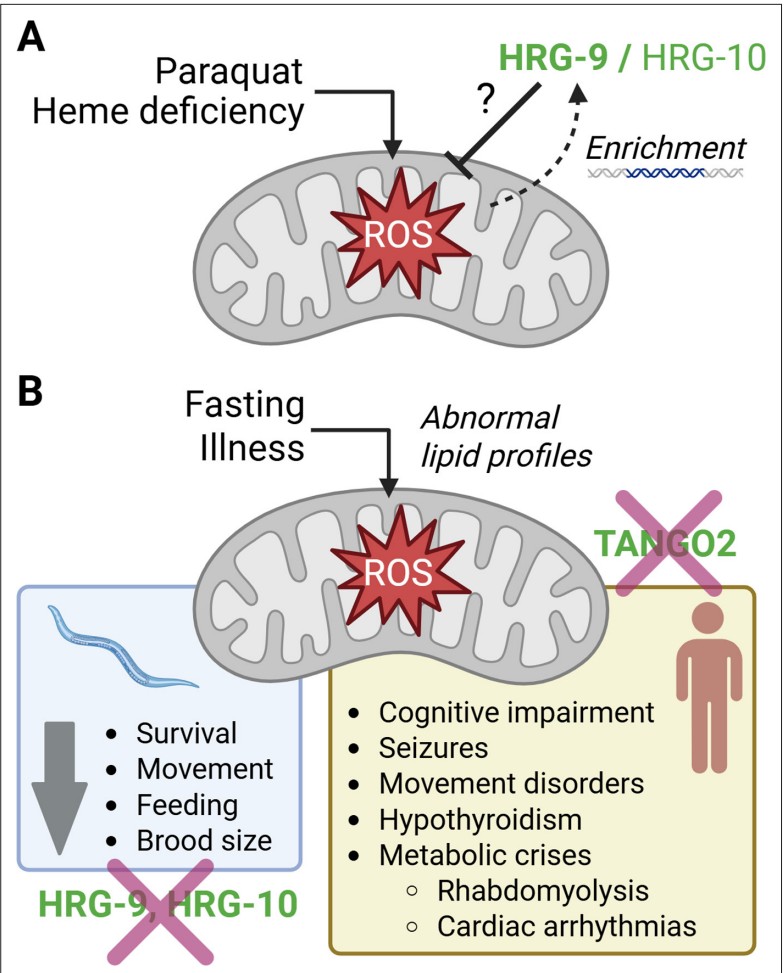

**Figure 5.** Proposed model of TANGO2 and its homologs acting as stress-responsive mediators of mitochondrial dysfunction. (**A**) Paraquat exposure and heme deficiency each induce reactive oxygen species (ROS) formation, mitochondrial stress, and enrichment of *hrg-9* in *C. elegans*. TANGO2 and its homologs may help mitigate mitochondrial stress under these conditions, though the exact function of these proteins remains unknown. (**B**) Physiological triggers such as fasting or illness also precipitate oxidative stress in the absence of TANGO2. Abnormal lipid profiles have been observed in multiple models of TANGO2 deficiency in the setting of impaired lipid mobilization and reduced fatty acid oxidation. In humans, loss of TANGO2 results in a complex clinical syndrome reminiscent of multiple secondary mitochondrial disorders. In *C. elegans*, loss of hrg-9 and hrg-10 induces a phenotype previously observed in nematode strains exhibiting mitochondrial dysfunction. Created with BioRender.com.

starvation (24 hours without OP50), and paraquat (25 mM), a potent generator of superoxide, with subsequent measurement of transcriptional enrichment by way of RT-qPCR. We observed a 12-fold enrichment of *hrg-9* after paraquat exposure, suggesting that its expression may be linked to oxidative stress more broadly and is not uniquely driven by heme levels (*Figure 3C*).

### Yeast deficient in TANGO2 homolog YGR127w exhibits normal growth

As Sun et al. also examined the function of TANGO2 homologs in yeast (YGR127w) and zebrafish (tango2), we sought to validate their findings in these models. It was previously reported that the *ygr127w* yeast knockout exhibits a severe temperature-sensitive growth defect and impaired heme distribution. We also observed a growth defect in the strain generated by Sun et al.; however, two separate strains on different backgrounds (BY4741 and BY4742) and a third strain generated by our lab (W303-1a) showed normal growth (*Figure 4A and B*). The growth defect in the initial yeast strain reported by Sun et al. was also not rescued with YGR127w complementation. As the background

for this strain is known to be prone to mitochondrial genome instability (*Dimitrov et al., 2009*), we hypothesize that this line may harbor a secondary mutation.

### Myofiber defects in tango2-deficient zebrafish

In zebrafish, Sun et al. reported no discernible phenotype in *tango2*[-/-] fish bred from heterozygous parents but observed severe skeletal muscle damage in *tango2*[-/-] larvae from *tango2*[-/-] parents. In a recent study, we showed increased lethality and reduced phospholipid and triglyceride levels in *tango2*[-/-] fish obtained from heterozygous parents (*Kim et al., 2023). tango2*[-/-] larvae from two different alleles (tango2[bwh210] and tango2[bwh211]) exhibited defects in myofiber organization irrespective of heterozygous or homozygous parents (*Figure 4C*) but lacked the striking muscle damage previously reported.

## Discussion

Our data, coupled with clinical observations in TDD, do not support a primary role for TANGO2 as a heme chaperone. In *C. elegans* lacking TANGO2 homologs, we were unable to fully replicate prior findings of defective heme transport using toxic and fluorescent heme analogs. Parallel studies in yeast and zebrafish failed to reproduce previously reported phenotypes in growth and muscle fiber integrity, respectively.

Exposing worms to GaPP, a toxic heme analog, we observed that nematodes deficient in HRG-9 and HRG-10 displayed increased survival compared to WT worms, consistent with prior work (*Sun et al., 2022*), though the between-group difference was markedly smaller in our study. We required higher GaPP concentrations to induce lethality, potentially due to product vendor differences, but did observe a clear dose-dependent effect across strains. Although it was previously proposed that the survival benefit seen in worms lacking HRG-9 and HRG-10 resulted from reduced transfer from intestinal cells after GaPP ingestion, our data suggest the reduced lethality is more likely due to decreased environmental GaPP uptake. Supporting this notion, DKO worms exhibited lawn avoidance, reduced pharyngeal pumping, and modestly lower intestinal ZnMP accumulation when exposed to this fluorescent heme analog on agar plates. In liquid media, DKO worms demonstrated higher fluorescence, but only in ZnMP-free conditions, suggesting the presence of gut granule autofluorescence. Furthermore, survival following exposure to GaPP was highest in *eat-2* mutants, despite heme trafficking being unaffected in this strain.

In addition to altered pharyngeal pumping, DKO worms displayed multiple previously unreported phenotypic features, suggesting a broader metabolic impairment reminiscent of some clinical manifestations observed in patients with TDD. Elucidating the mechanisms underlying this phenotype, and whether they reflect a core bioenergetic defect, is an active area of investigation in our lab. Several *C. elegans* heme-responsive genes have been characterized, revealing relatively specific defects in heme uptake or utilization rather than broad organismal dysfunction. For example, *hrg-1* and *hrg-4* mutants exhibit impaired growth only under heme-limited conditions (*Yuan et al., 2012*), and *hrg-3* loss affects brood size and embryonic viability specifically when maternal heme is scarce (*Chen et al., 2011*). By contrast, *hrg-9* and *hrg-10* mutants exhibit the most severe phenotypes of the hrg family to date, including reduced pharyngeal pumping, decreased motility, shortened lifespan, and smaller broods, even when fed a heme-replete diet.

Laboratory abnormalities in human patients TDD include abnormal acylcarnitine profiles, hyperammonemia, and elevated creatinine kinase levels during metabolic crises (*Powell et al., 2021*; *Miyake et al., 2023*), while abnormalities associated with defective heme transport (e.g., erythrocyte membrane defects, low hemoglobin levels) have not been observed in patients with this condition. Strikingly, retrospective data suggest that patients with TDD receiving B-vitamin supplementation inclusive of pantothenic acid, a precursor of CoA, do not experience metabolic crises (*Miyake et al., 2023*) and show substantial improvement in other domains as well. Pantothenic acid supplementation also yielded full phenotypic rescue in a *Drosophila* model of TDD (*Asadi et al., 2023*). It is difficult to reconcile how this treatment would be beneficial in a condition characterized by dysregulated heme trafficking.

Heme is a hydrophobic molecule; thus, it is plausible that if TANGO2 and its homologs are involved in lipid binding, as was recently demonstrated in Hep2G cells (*Lujan et al., 2025*), these proteins may

also weakly bind heme. *Han et al., 2023* demonstrated that a bacterial heme homolog, HtpA, directly binds heme and is necessary for cytochrome c function. We would note, however, that TANGO2 was not among the 378 heme-binding proteins identified on a recent proteomic screen of three separate cell lines (*Homan et al., 2022*). *Jayaram et al., 2025* recently proposed that TANGO2 may instead interact with the mitochondrial heme exporter FLVCR1b to release mitochondrial heme without directly binding to heme itself, though this interaction was observed only after heme synthesis was potentiated via d-ALA and iron supplementation, raising questions about the role of TANGO2 under basal heme conditions. Furthermore, the investigators identified GAPDH as the exclusive binding partner of heme upon export through FLVCR1b. How GAPDH, a protein important for multiple cellular functions, including glycolysis, is affected in TDD remains unknown.

In this study, we demonstrated that *hrg-9* expression is strongly induced by paraquat, a generator of superoxide free radical. Prior work has also identified *hrg-9* as a major transcriptional target of the mitochondrial unfolded protein response (mtUPR) through ATFS-1 transcription factor binding (*Soo and Van Raamsdonk, 2021*; *Di Pede et al., 2025*). Given that heme is an essential component of cytochromes in the electron transport chain, heme deficiency could plausibly activate *hrg-9* through the induction of mitochondrial stress. Conversely, artificially stimulating heme synthesis may exert a similar transcriptional effect, as excess heme is mitotoxic and can itself trigger a stress response. Indeed, *hrg-9* was also shown to be modestly upregulated in the high-heme state, unlike other genes in the *hrg* family (*Figure 3A*), and our reanalysis of an RNA-seq dataset examining transcription under low-heme conditions revealed broad induction of stress-responsive genes with no established role in heme trafficking (*Figure 3B*).

In summary, our findings challenge the notion that TANGO2 functions as a heme chaperone. Instead, and consistent with growing clinical evidence, they support a model in which TANGO2 may help mitigate cellular stress and maintain mitochondrial function under conditions of redox imbalance (*Figure 5*). Clearly, further work is needed to define the precise role of this highly conserved protein as we work to develop effective treatments for patients with TDD.

# Materials and methods

**Key resources table**

| Reagent type (species) or resource | Designation | Source or reference | Identifiers | Additional information |
|---|---|---|---|---|
| Strain, strain background (*Caenorhabditis elegans*) | N2 | Caenorhabditis Genetics Center (CGC) at the University of Minnesota | Bristol N2 | |
| Strain, strain background (*C. elegans*) | Double knockout (DKO) | Chen Lab, Zhejiang University | CCH303 (hrg-9(cck301)V; hrg-10(cck302)V) | *Sun et al., 2022* |
| Strain, strain background (*C. elegans*) | eat-2 | Samuelson Lab, University of Rochester | eat-2(ad465) | |
| Chemical compound, reagent | Gallium protoporphyrin IX chloride (GaPP) | Santa Cruz | CAS 210409-12-4 | GaPP used in *Sun et al., 2022* sourced from Frontier Scientific |
| Chemical compound, reagent | Zn(II) Mesoporphyrin IX (ZnMP) | Santa Cruz | CAS 14354-67-7 | ZnMP used in *Sun et al., 2022* sourced from Frontier Scientific |
| Software, algorithm | Empirical Analysis of Gene Expression in R (edgeR) | Bioconductor.org | | *Sun et al., 2022* |
| Software, algorithm | WormCat | Github.com | | *Holdorf et al., 2020* |
| Chemical compound, reagent | Paraquat (methyl viologen dichloride hydrate) | Sigma-Aldrich | 856177 | |

*Continued on next page*

*Continued*

| Reagent type (species) or resource | Designation | Source or reference | Identifiers | Additional information |
|---|---|---|---|---|
| Chemical compound, reagent | TRIzol | Invitrogen | 15596026 | |
| Commercial assay or kit | RNeasy Mini Kit | QIAGEN | 74104 | |
| Commercial assay or kit | Maxima First Strand cDNA Synthesis Kit | Thermo Fisher | K1671 | |
| Chemical compound, reagent | iTaq Universal SYBR Green Supermix | Invitrogen | 1725121 | |
| Recombinant DNA reagent | *hrg-9* F | This paper | PCR primers | GGACCCGCTGCCATACACTAATC |
| Recombinant DNA reagent | *hrg-9* R | This paper | PCR primers | GACAATTCAAATCTGGCATCGTG |
| Recombinant DNA reagent | *hrg-10* F | This paper | PCR primers | AGGCTTCCCGGAGCACATTTAC |
| Recombinant DNA reagent | *hrg-10* R | This paper | PCR primers | CAGGCTCCATGCGTCTATCCAG |
| Recombinant DNA reagent | *act* F | This paper | PCR primers | CAACACTGTTCTTTCCGGAG |
| Recombinant DNA reagent | *act* R | This paper | PCR primers | CTTGATCTTCATGGTTGATGGG |
| Gene (*Danio rerio*) | *tango2* | ENSMBL | ENSDARG00000056550 | Zebrafish homolog of TANGO2 |
| Strain, strain background (*D. rerio*) | TUAB | Zebrafish International Resource Center | TU (ZL57) AB (ZL1) | Wild-type lines. Sex is not determined in *Danio rerio* at the age group animals used in this study |
| Genetic reagent (*D. rerio*) | sgRNA to *tango2* | Thermofisher Scientific; **Kim et al., 2023** | | Guide RNA to create mutations in tango2 gene |
| Sequence-based reagent | *tango2* F | **Kim et al., 2023** | PCR primer | ATGGCTGAAAGAGCTGTGCT |
| Sequence-based reagent | tango2 R | **Kim et al., 2023** | PCR Primer | ATGGCTGAAAGAGCTGTGCT |
| Chemical compound, drug | Alexa Fluor 488-Phalloidin | Thermo Fisher Scientific | A12379 | 1:40 dilution |
| Chemical compound, drug | Methylcellulose | Sigma-Aldrich | M0387-500G | 1% w/v dilution |
| Software, algorithm | Prism | GraphPad | | |
| Special instrumentation | Zebrafish automated activity monitor | Zantiks | ZantiksMWP | |
| Gene (*Saccharomyces cerevisiae*) | ygr127w | Saccharomyces Genome Database (SGD) | S000003359 | Yeast homolog of human Tango2 |
| Strain, strain background: (*S. cerevisiae*, haploid, MATa) | BY4741 (MATa his3Δ1 leu2Δ0 met15Δ0 ura3Δ0) | EUROSCARF | | |
| Strain, strain background: (*S. cerevisiae*,haploid, MATα) | BY4742 (his3Δ1 leu2Δ0 lys2Δ0 ura3Δ0) | EUROSCARF | | |

*Continued on next page*

*Continued*

| Reagent type (species) or resource | Designation | Source or reference | Identifiers | Additional information |
|---|---|---|---|---|
| Strain, strain background: (*S. cerevisiae*, haploid, MATa) | W301 (leu2-3,112 trp1-1 can1-100 ura3-1 ade2-1 his3-11,15) | Lab stock | | |
| Genetic reagent (*S. cerevisiae*) | YGR127W::KanMX in BY4741, BY4742 | Lab stock, Chen Lab, Zhejiang University | YGR127W::KanMX | *Sun et al., 2022* |
| Genetic reagent (*S. cerevisiae*) | YGR127W::KanMX in W303 | This paper | YGR127W::KanMX | Deletion strain generated by PCR replacement. verified by locus-specific PCR |
| Genetic reagent (*S. cerevisiae*) | sgRNA: *ygr127w* | Integrated DNA Technologies (IDT) | | Guide RNA sequence used for CRISPR-Cas9–mediated deletion of YGR127W; replaced with KanMX resistance cassette to generate ygr127wΔ strain |
| Recombinant DNA reagent | pRS313-YGR127W (HIS3, CEN6) | This paper | | Yeast centromeric plasmid expressing YGR127W used for complementation |
| Sequence-based reagent | ygr127wgRNA-F | This paper | sgRNA: *ygr127w* | GACTTTAATACGAAATCAAGATCCCG |
| Sequence-based reagent | ygr127wgRNA-R | This paper | sgRNA: *ygr127w* | AAACCGGGATCTTGATTTCGTATTAA |
| Sequence-based reagent | ygr127 F | This paper | PCR primer | AGTCGGATCCTCAAGGTTCTTCTCTAGAACC |
| Sequence-based reagent | ygr127 R | This paper | PCR primer | AGTCGGATCCTGCTCTATTGGAGAACTTAACC |
| Sequence-based reagent | ygr127 F | This paper | PCR primer | TTGGCATCTGCCTAGCTTTCG |
| Sequence-based reagent | ygr127 R | This paper | PCR primer | AGCGTCTACTGTGGTTACTG |
| Commercial assay or kit | Q5 High-Fidelity DNA Polymerase | New England Biolabs (NEB) | | Cloning of inserts into pRS313 backbones |
| commercial assay or kit | T4 DNA Ligase | New England Biolabs (NEB) | | Cloning of inserts into pRS313 backbones |
| Commercial assay or kit | CutSmart Buffer and Restriction Enzymes (EcoRI, XhoI, BamHI) | New England Biolabs (NEB) | | Cloning of inserts into pRS313 backbones |
| Commercial assay or kit | Golden Gate Assembly Kit (Type IIS cloning) | New England Biolabs (NEB) | | Used for one-pot modular assembly of plasmid constructs |
| Commercial assay or kit | EZ Yeast Transformation Kit | Zymo Research | Used for high-efficiency yeast transformation during plasmid integration and complementation assays | |

## Worm strains and maintenance

Worm strains used include Bristol N2 (obtained from the Caenorhabditis Genetics Center [CGC] at the University of Minnesota), DKO (CCH303 (hrg-9(cck301)V; hrg-10(cck302)V) obtained from C. Chen, and eat-2(ad465)), obtained from the Samuelson lab at the University of Rochester. All worms were maintained at 20°C on standard nematode growth medium (NGM) plates with OP50 *E. coli*. All *C. elegans* assays were performed and scored by blinded observers.

## GaPP survival assay

Standard OP50 NGM plates treated with 1, 2, 5, or 10 µM gallium protoporphyrin IX (GaPP; Santa Cruz) after seeding. Plates were swirled to ensure an even distribution of GaPP and allowed to dry

completely. Adult worms were placed on plates, permitted to lay eggs for 24 hours and were then removed. Offspring were assessed 72 hours later for survival and were scored as dead if they did not respond to prodding with a platinum wire worm pick.

## Pharyngeal pumping
L4 worms were observed and video recorded for 60-second intervals. The number of pharyngeal pumps per minute was manually counted.

## Lawn avoidance
L4 worms were placed in the center of OP50 *E. coli* lawns on standard NGM plates and incubated at 20°C for 24 hours. Plates were examined 24 hours later, and the numbers of worms remaining on the lawn or on unseeded agar were counted. Worms found on the sides of the dish or otherwise absent from NGM surface were scored as off the plate.

## Brood size
For 24-hour brood assessments, young adult worms were placed on plates and allowed to lay eggs for 24 hours before being removed. Viable offspring were counted 48 hours later. For total brood assessments, L4 worms were placed on plates and moved to fresh plates every 24 hours for 5 days to ensure all eggs were accounted for. Viable offspring from eggs laid on each plate were counted 48 hours later and summed.

## ZnMP fluorescence
L4 worms grown on OP50 plates were placed on plates treated with 40 uM zinc mesoporphyrin (ZnMP; Santa Cruz) for 16 hours before being washed in M9 buffer, anesthetized in sodium azide, and mounted on slides with 2% agarose pads for imaging. Worms in the starved condition were placed on unneeded plates for 16 hours before preparation and mounting as above. For liquid media experiments, three generations of worms were cultured in regular heme (20 uM) axenic media, with the first two generations receiving antibiotic-supplemented media (10 mg/ml tetracycline) and the third generation cultivated without antibiotic. L4 worms from the third generation were placed in media containing 40 uM ZnMP for 16 hours before being prepared and mounted for imaging as above. Worms were imaged on Zeiss Axio Imager 2 at ×40 magnification, with image settings kept uniform across all images. Fluorescent intensity was measured within the proximal region of the intestine using ImageJ.

## RNA-seq cluster and gene ontology analysis
The RNA sequencing (RNA-seq) dataset generated by Sun et al. was analyzed with Empirical Analysis of Gene Expression in R (edgeR) and the top 500 genes were extracted. Computational cluster analysis was done in RStudio. Gene ontology analysis was performed on significantly enriched genes using WormCat (*Holdorf et al., 2020*). Source code is available in supplemental materials.

## Stress conditions and RT-qPCR
N2 worms were subjected to the following stress conditions prior to RNA extraction: (1) fasting: worms were deprived of OP50 *E. coli* for 24 hours; (2) heat: worms were incubated for 4 hours at 34°C; and (3) paraquat: worms were placed on standard NGM/OP50 plates treated with 25 mM paraquat (methyl viologen dichloride hydrate; Sigma Aldrich) for 24 hours. Whole worm RNA was extracted with TRIzol (Invitrogen) and purified using the RNeasy Mini Kit (QIAGEN). cDNA generation was performed with the Maxima First Strand cDNA Synthesis Kit (Thermo Fisher). qPCR was performed in triplicate using iTaq Universal SYBR Green Supermix (Invitrogen) with a CFX Duet Realtime qPCR machine (Bio-Rad). Gene expression was normalized to worm *act* and analyzed using the ΔΔCq method. Primers used were as follows: *hrg-9*: GGACCCGCTGCCATACACTAATC and GACAATTCAAATCTGGCATCGTG *hrg-10*: AGGCTTCCCGGAGCACATTTAC and CAGGCTCCATGCGTCTATCCAG
    *act*: CAACACTGTTCTTTCCGGAG and CTTGATCTTCATGGTTGATGGG

## Yeast strain construction

*YGR127w* was knocked out by homologous recombination. A strain (BY4741) harboring a *ygr127w*D was used as a template and the knockout cassette was amplified by PCR using the following primers that anneal upstream and downstream of the locus: TTGGCATCTGCCTAGCTTTCG and AGCGTCTACTGTGGTTACTG

The PCR amplicon was then transformed into W303-1a cells and transformants were selected on YPD plates containing 200 mg/ml G418. Integration at the correct locus was confirmed by PCR using the same primers, as above. To complement the *ygr127w*D in the Sun et al. strain, wild type *YGR127w* was amplified with 400 base pairs on either side of the gene and cloned into a *HIS3*-containing plasmid (pRS413). Transformants were selected on synthetic complete (SC) medium lacking histidine.

## Yeast growth curves

Cells were grown to stationary phase at 25°C in either SC or SC-histidine medium. Cultures in the same medium were inoculated at an OD600 of ~0.01 in a 100 ml volume in a 96-well plate. The OD600 was read every 15 minutes on a Sunrise Tecan microplate reader.

## Zebrafish

All procedures involving zebrafish were approved by the Brigham and Women's Hospital Animal Care and Use Committee. Fish were bred and maintained using standard methods as described (*Westerfield, 2000*). tango2bwh210 and tango2bwh211 zebrafish lines were created in our laboratory by the CRISPR-Cas9 approach as described previously (*Kim et al., 2023*). The tango2bwg210 allele has an insertion of seven bases (c.226_227ins7; p.Tyr76Leufs*25) and tango2bwg211 has a 26 base insertion in exon2 (c.226_227ins26; p.Tyr76Leufs*207) resulting in frameshift mutations and loss of protein function. Whole mount phalloidin staining and microscopy was performed as described previously (*Casey et al., 2023*).

## Statistics and reproducibility

All statistical analyses were performed in GraphPad Prism 9. Data presented are mean ± S.E.M. One-way analysis of variance followed by Bonferroni's multiple comparisons was used to determine statistical significance ($p < 0.05$). Sample sizes were not predetermined.

# Acknowledgements

The authors would like to thank Dr. Keith Nehrke and Dr. Paul Brookes for their guidance and manuscript review, the Chen lab for providing the hrg-9/hrg-10 knockout nematode strain, and the Samuelson lab for supplying the eat-2 nematode strain. SES is a trainee in the Medical Scientist Training Program funded by NIH T32 GM007256. SJM is supported by the National Institutes of Health (NIH) National Institute of Neurological Disorders and Stroke Grant #2K12NS098482-06. This work was funded in part by grants from the TANGO2 Research Foundation to VG, MS, SJM, and LW.

# Additional information

### Competing interests

Sarah E Sandkuhler: has an unpaid role at the TANGO2 Research Foundation - Early Diagnostic and Detection Committee Member. Vandana Gupta: has an unpaid role at the TANGO2 Research Foundation - Scientific Advisor Board Member. Michael Sacher: has an unpaid role at the TANGO2 Research Foundation - Scientific Advisory Board Member, Executive Board Member. Samuel J Mackenzie: has an unpaid role at the TANGO2 Research Foundation - Advisory Board Chair, Executive Board Member. The other authors declare that no competing interests exist.

### Funding

| Funder | Grant reference number | Author |
| --- | --- | --- |
| NIH | T32 GM007256 | Sarah E Sandkuhler |

| Funder | Grant reference number | Author |
| --- | --- | --- |
| TANGO2 Research Foundation | | Lili Wang<br>Vandana Gupta<br>Michael Sacher<br>Samuel J Mackenzie |
| National Institute of Neurological Disorders and Stroke | 2K12NS098482-06 | Samuel J Mackenzie |

The funders had no role in study design, data collection and interpretation, or the decision to submit the work for publication.

## Author contributions

Sarah E Sandkuhler, Conceptualization, Data curation, Formal analysis, Investigation, Methodology, Writing – original draft, Writing – review and editing; Kayla S Youngs, Conceptualization, Data curation, Investigation, Methodology; Olivia Gottipalli, Data curation, Formal analysis, Investigation; Laura D Owlett, Data curation, Software, Formal analysis, Methodology; Monica B Bandora, Investigation; Aaliya Naaz, Euri Kim, Data curation, Investigation; Lili Wang, Conceptualization, Funding acquisition, Visualization; Andrew Wojtovich, Conceptualization, Methodology, Writing – review and editing; Vandana Gupta, Michael Sacher, Conceptualization, Supervision, Funding acquisition, Visualization, Methodology, Writing – review and editing; Samuel J Mackenzie, Conceptualization, Resources, Data curation, Software, Formal analysis, Supervision, Funding acquisition, Validation, Investigation, Visualization, Methodology, Writing – original draft, Project administration, Writing – review and editing

## Author ORCIDs

Sarah E Sandkuhler https://orcid.org/0000-0001-6799-9566
Lili Wang https://orcid.org/0000-0002-2746-4521
Vandana Gupta https://orcid.org/0000-0002-4057-8451
Michael Sacher https://orcid.org/0000-0003-2926-5064
Samuel J Mackenzie https://orcid.org/0000-0002-5571-7370

## Ethics

All procedures involving zebrafish were approved by the Brigham and Women's Hospital Animal Care and Use Committee.

Reviewer #1 (Public review): https://doi.org/10.7554/eLife.105418.3.sa1
Reviewer #2 (Public review): https://doi.org/10.7554/eLife.105418.3.sa2
Reviewer #3 (Public review): https://doi.org/10.7554/eLife.105418.3.sa3
Author response https://doi.org/10.7554/eLife.105418.3.sa4

# Additional files

## Supplementary files

MDAR checklist

Supplementary file 1. Raw data from phenotypic assays and RT-qPCR.

Source code 1. R code used for analysis of differential gene expression and hierarchical clustering.

## Data availability

All raw data from worm behavior, yeast, zebrafish, and RNA-seq studies are available in *Supplementary file 1*.

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
