## [Editor Report · eLife Assessment]

This **valuable** study provides **solid** evidence that supports TANGO2 homologs, including HRG-9 and HRG-10, can play a role in cellular bioenergetics and oxidative stress homeostasis. It also challenges the previously reported role of TANGO2 in heme transport and paves the way for future mechanistic studies addressing the mechanisms of how TANGO2 regulates oxidative stress homeostasis. The strengths include the use of different model systems, genetic tools, behavioral assays, and efforts by the authors in using the same reagents to reproduce results of other groups.

---

## [Referee Report · Reviewer #1 (Public review)]

Sandkuhler et al. re-evaluated the biological functions of TANGO2 homologs in *C. elegans*, yeast, and zebrafish. Compared to the previously reported role of TANGO2 homologs in transporting heme, Sandkuhler et al. expressed a different opinion on the biological functions of TANGO2 homologs. With the support of some results from their tests, they conclude that 'there is insufficient evidence to support heme transport as the primary function of TANGO2', in addition to the evidence that *C. elegans* TANGO2 helps counteract oxidative stress.. While the differences are reported in this study, more work is needed to elucidate the intuitive biological function of TANGO2.

Strengths:

(1) This work revisits a set of key experiments, including the toxic heme analog GaPP survival assay, the fluorescent ZnMP accumulation assay, and the multi-organismal investigations documented by Sun et al. in Nature (2022), which are critical for comparing the two works. Meanwhile, the authors also highlight the differences in reagents and methods between the two studies, demonstrating significant academic merit.

(2) This work reported additional phenotypes for the *C. elegans* mutant of the TANGO2 homologs, including lawn avoidance, reduced pharyngeal pumping, smaller brood size, faster exhaustion under swimming test, and a shorter lifespan. These phenotypes are important for understanding the biological function of TANGO2 homologs, while they were missing from the report by Sun et al.

(3) Investigating the 'reduced GaPP consumption' as a cause of increased resistance against the toxic GaPP for the TANGO2 homologs, hrg-9 hrg-10 double null mutant provides a valuable perspective for studying the biological function of TANGO2 homologs.

(4) The induction of hrg-9 gene expression by paraquat indicates a strong link between TANGO2 and mitochondrial function.

(5) This work thoroughly evaluated the role of TANGO2 homologs in supporting yeast growth using multiple yeast strains and also pointed out the mitochondrial genome instability feature of the yeast strain used by Sun et al.

Weakness:

It is always a challenge to replicate someone else's work, but it is worthwhile to take on the challenge, provide evidence, and raise concerns about it. These authors attempted to replicate the experiment using the same biological material as that used by Sun et al. in Nature (2022), despite some experimental differences between the two studies. This study does not have many technical weaknesses, but it can become a much better project by focusing on the new phenotypes discovered here.

---

## [Referee Report · Reviewer #2 (Public review)]

This work offers a valuable re-evaluation of earlier claims from other groups about TANGO2 functions and proposes that energy-related and stress-related pathways may be more important to the disorder than previously thought. A key strength of this work is the use of multiple model systems. The authors provide solid data that show how TANGO2 is probably only indirectly involved in heme transport and provide support for alternative mechanisms where TANGO2 is actually directly control. These findings provide valuable information for researchers seeking more accurate therapeutic targets.

Strengths:

The study refutes earlier claims about TANGO2's involvement in heme transport and extends previous findings by implicating TANGO2 in metabolism and oxidative stress, thereby highlighting new aspects of its role in cell physiology. The use of different model systems (*Saccharomyces cerevisiae*, *Caenorhabditis elegans*, *Danio rerio*) to address the main research questions is useful and demonstrates evolutionary conservation of the studied processes. Finally, the results suggest a broader impact than previously described, somewhat supporting the novelty of the study.

Weaknesses:

Although the phenotypic analyses are broad and generally well executed, a key limitation is that the main conclusions mainly rely on these readouts. While informative, sole phenotypic analyses cannot directly demonstrate the underlying molecular mechanisms proposed by the authors. The study includes limited functional or biochemical assays connecting TANGO2 orthologs to the proposed energy and stress pathways. Some observations would benefit from additional orthogonal validation to strengthen the overall interpretation. As a result, the evidence supporting the central mechanistic interpretation remains indirect, although compelling.

Overall, the authors have achieved their stated aims, and their results mainly support their main conclusion (i.e., TANGO2 is unlikely to function in heme transport and is probably linked to energy and stress pathways). However, much of the evidence comes from phenotypic analyses, which limits the strength of the mechanistic claims, leaving the proposed pathways somewhat indirect.

This work is likely to have a valuable impact on the subfield by clarifying that TANGO2 is not involved (at least directly) in heme transport and clarifying its actual role in energy and stress-related processes. By rigorously reassessing and confuting earlier claims from other studies across multiple model systems, the current work will help to guide the future research and therapeutic exploration in the context of TANGO2 deficiencies. This study will provide a solid foundation for more mechanistic insights into TANGO2 function.

---

## [Referee Report · Reviewer #3 (Public review)]

In this paper, Sandkuhler et al. reassessed the role of TANGO2 as a heme chaperone proposed by Sun et al in a recently published paper (https://doi.org/10.1038/s41586-022-05347-z). Overall, Sandkuhler et al. conclude that the heme-related roles of TANGO2 had been overemphasized by Sun et al. especially because the hrg9 gene does not exclusively respond to different regimens of heme synthesis/uptake but is susceptible to a greater extent to, for example, oxidative stress. Impaired heme trafficking is then interpreted as due to general mitochondrial dysfunction. In recent years, the discussion around the heme-related roles of TANGO2 has been tantalizing but is still far from a definitive consensus. Discrepancies between results and their interpretation are testament to how ambitious the understanding of TANGO2 and the phenotypes associated with TANGO2 defects are.

The work presented by Sandkuhler et al. is methodologically sound, and the authors have appropriately addressed my concerns in the first round of review. Overall, this paper challenges the recent developments in the field in relation to heme trafficking and provides a wider perspective on the biological roles of TANGO2.

---

## [Author Response]

The following is the authors’ response to the original reviews.

**Reviewer #1 (Public review):**
(1) A detailed comparison between this work and the work of Sun et al. on experimental protocols and reagents in the main text will be beneficial for readers to assess critically.

We have added a Key Reagents Table outlining the key reagents used in our study. In terms of experimental protocols, we replicated those described by Sun et al. in most instances and described any differences when present. With this resubmission, we included additional ZnMP accumulation experiments in liquid media (see point 3 below).

(2) The GaPP used by Sun et al. (purchased from Frontier Scientific) is more effective in killing the worm than the one used in this study (purchased from Santa Cruz). Is the different outcome due to the differences in reagents? Moreover, Sun et al. examined the lethality after 3-4 days, while this work examined the lethality after 72 hours. Would the extra 24 hours make any difference in the result?

We now cite product vender differences as a possible reason for the observed difference in worm death, as the reviewer suggests, on page 8 (see text below) and include these differences in the Key Reagents Table. We also now stress the fact that our experiments included different doses of GaPP and the use of eat-2 mutants as an additional control, which we believe adds rigor and demonstrates the potency of GaPP in our experiments. We decided on assessment at 72 hours, as we deemed it a less nebulous time point as compared to 3-4 days. Most of the observed worm death occurred earlier in this interval, so we believe it is unlikely that large group differences would emerge after an additional 24 hours.

“Exposing worms to GaPP, a toxic heme analog, we observed that nematodes deficient in HRG-9 and HRG-10 displayed increased survival compared to WT worms, consistent with prior work,[13] though the between-group difference was markedly smaller in our study. We required higher GaPP concentrations to induce lethality, potentially due to product vendor differences, but did observe a clear dose-dependent effect across strains. Although it was previously proposed that the survival benefit seen in worms lacking HRG-9 and HRG-10 resulted from reduced transfer from intestinal cells after GaPP ingestion, our data suggest the reduced lethality is more likely due to decreased environmental GaPP uptake. Supporting this notion, DKO worms exhibited lawn avoidance, reduced pharyngeal pumping, and modestly lower intestinal ZnMP accumulation when exposed to this fluorescent heme analog on agar plates. In liquid media, DKO worms demonstrated higher fluorescence, but only in ZnMP-free conditions, suggesting the presence of gut granule autofluorescence. Furthermore, survival following exposure to GaPP was highest in eat-2 mutants, despite heme trafficking being unaffected in this strain.”

(3) This work reported the opposite result of Sun et al. for the fluorescent ZnMP accumulation assay. However, the experimental protocols used by the two studies are massively different. Sun et al. did the ZnMP staining by incubating the L4-stage worms in an axenic mCeHR2 medium containing 40 μM ZnMP (purchased from Frontier Scientific) and 4 μM heme at 20 ℃ for 16 h, while this work placed the L4-stage worms on the OP50 *E. coli* seeded NGM plates treated with 40 μM ZnMP (purchased from Santa Cruz) for 16 h. The liquid axenic mCeHR2 medium is bacteria-free, heme-free, and consistent for ZnMP uptake by worms. This work has mentioned that the hrg-9 hrg-10 double null mutant has bacterial lawn avoidance and reduced pharyngeal pumping phenotypes. Therefore, the ZnMP staining protocol used in this work faces challenges in the environmental control for the wild type vs. the mutant. The authors should adopt the ZnMP staining protocol used by Sun et al. for a proper evaluation of fluorescent ZnMP accumulation.

We agree with this comment. As such, we performed the ZnMP assay in liquid media conditions, as now described on page 13:

“For liquid media experiments, three generations of worms were cultured in regular heme (20 uM) axenic media, with the first two generations receiving antibiotic-supplemented media (10 mg/ml tetracycline) and the 3^rd^ generation cultivated without antibiotic. L4 worms from the 3^rd^ generation were placed in media containing 40uM ZnMP for 16 hours before being prepared and mounted for imaging as above. Worms were imaged on Zeiss Axio Imager 2 at 40x magnification, with image settings kept uniform across all images. Fluorescent intensity was measured within the proximal region of the intestine using ImageJ.”

In heme-free media, both WT and DKO worms invariably entered L1 arrest, thus we were not able to replicate the results reported by Sun et al. Using media containing heme, we did see an increase in fluorescence, but this was only in the ZnMP-free condition, indicating that the increased signal was attributable to autofluorescence. This is a known phenomenon associated with gut granules in *C. elegans* in the setting of oxidative stress. The results of these experiments are now summarized on page 6:

“DKO nematodes at the L4 larval stage were previously shown to accumulate the fluorescent heme analog zinc mesoporphyrin IX (ZnMP) in intestinal cells in low-heme (4 µM) liquid media. While attempting to replicate this experiment, we observed that both wildtype and DKO nematodes entered L1 arrest under these conditions. Therefore, to allow for developmental progression, we grew worms on standard OP50 *E. coli* plates and in media containing physiological levels of heme (20 µM). We then examined whether differences in ZnMP uptake persisted under these basal conditions. DKO worms grown on ZnMP-treated *E. coli* plates displayed significantly reduced intestinal ZnMP fluorescence compared to N2 (Figure 1B and C). Using basal heme media with ZnMP, there was no significant difference in ZnMP fluorescence between DKO and wildtype nematodes, although DKO worms grown in media without ZnMP exhibited significantly higher autofluorescence (Figure 1D and E). To test whether autofluorescence may have contributed to the higher fluorescent intensities previously reported in heme-deficient DKO worms, we repeated this experiment on agar plates under starved conditions but did not observe a difference between groups (Figure 1B).”

(4) A striking difference between the two studies is that Sun et al. emphasize the biochemical function of TANGO2 homologs in heme transporting with evidence from some biochemical tests. In contrast, this work emphasizes the physiological function of TANGO2 homologs with evidence from multiple phenotypical observations. In the discussion part, the authors should address whether these observed phenotypes in this study can be due to the loss of heme transporting activities upon eliminating TANGO2 homologs. This action can improve the merit of academic debate and collaboration.

Thank you for this suggestion. The following text has been added to the Discussion section (page 9):

“In addition to altered pharyngeal pumping, DKO worms displayed multiple previously unreported phenotypic features, suggesting a broader metabolic impairment and reminiscent of some clinical manifestations observed in patients with TDD. Elucidating the mechanisms underlying this phenotype, and whether they reflect a core bioenergetic defect, is an active area of investigation in our lab. Several *C. elegans* heme-responsive genes have been characterized, revealing relatively specific defects in heme uptake or utilization rather than broad organismal dysfunction. For example, hrg-1 and hrg-4 mutants exhibit impaired growth only under heme-limited conditions,[23] and hrg-3 loss affects brood size and embryonic viability specifically when maternal heme is scarce.[24] By contrast, hrg-9 and hrg-10 mutants exhibit the most severe organismal phenotypes of the hrg family, to date, including reduced pharyngeal pumping, decreased motility, shortened lifespan, and smaller broods, even when fed a heme-replete diet.”

**Reviewer #2 (Public review):**
(1) The manuscript is written mainly as a criticism of a previously published paper. Although reproducibility in science is an issue that needs to be acknowledged, a manuscript should focus on the new data and the experiments that can better prove and strengthen the new claims.

Thank you for this suggestion. While the primary intent of this study was to replicate key findings from the 2022 publication by Sun et al., the revised manuscript now emphasizes underlying mechanisms more broadly rather than focusing narrowly on that prior publication.

(2) The current presentation of the logic of the study and its results does not help the authors deliver their message, although they possess great potential.

We have attempted to rectify this through substantial revision of the Discussion section and other places throughout the manuscript.

(3) The study is missing experiments to link hrg-9 and hrg-10 more directly to bioenergetic and oxidative stress pathways.

The reviewer is correct in this assertion, but it was not our intent to definitively prove this link or, indeed, the primary mechanism of TANGO2 in the present manuscript. This said, we are actively engaged in this endeavor in our lab and anticipate these data will be published in a separate, forthcoming publication.

We have added additional references pertaining to hrg-9 enrichment as part of the mitochondrial unfolded protein response (page 10) and a comparison of the phenotype observed in hrg-9 and hrg-10 deficient worms versus those lacking other proteins in the hrg family (page 9).

**Reviewer #3 (Public review):**
(1) The authors stress - with evidence provided in this paper or indicated in the literature - that the primary role of TANGO2 and its homologues is unlikely to be related to heme trafficking, arguing that observed effects on heme transport are instead downstream consequences of aberrant cellular metabolism. But in light of a mounting body of evidence (referenced by the authors) connecting more or less directly TANGO2 to heme trafficking and mobilization, it is recommended that the authors comment on how they think TANGO2 could relate to and be essential for heme trafficking, albeit in a secondary, moonlighting capacity. This would highlight a seemingly common theme in emerging key players in intracellular heme trafficking, as it appears to be the case for GAPDH - with accumulating evidence of this glycolytic enzyme being critical for heme delivery to several downstream proteins.

TANGO2 is essential for mitochondrial health, albeit in a yet unknown capacity. In the absence of TANGO2, defects in heme trafficking may be secondary sequelae of mitochondrial dysfunction. We would point out that prior studies that attempted to show that TANGO2 and its homologs are involved in heme trafficking proposed very different mechanisms (direct binding vs. membrane protein interaction) and relied on artificially low or high heme conditions to produce these effects. We have attempted to address these more clearly in the Discussion section and have added a fifth figure to summarize our current unifying theory for how heme levels and mitochondrial stress may be linked.

(2) The observation - using eat-2 mutants and lawn avoidance behaviour - that survival patterns can be partially explained by reduced consumption, is fascinating. It would be interesting to quantify the two relative contributions.

We have completed additional ZnMP experiments in liquid media at the reviewers’ request. This experimental condition eliminates lawn avoidance as a factor in consumption. Fluorescent intensity was significantly higher in the DKO worms in media lacking ZnMP, indicating increased autofluorescence in DKO worms, while signal was not significantly different in media with ZnMP.

(3) In the legend to Figure 1A it's a bit unclear what the differently coloured dots represent for each condition. Repeated measurements, worms, independent experiments? The authors should clarify this.

The following sentence has been added to the legend for Figure 1:

“Each dot represents the number of offspring laid by one adult worm on one GaPP-treated plate after 24 hours.”

(4) It would help if the entire fluorescence images (raw and processed) for the ZnMP treatments were provided. Fluorescence images would also benefit Figure 1B.

Fluorescent intensity values pertaining to the ZnMP experiments are included in our Extended Data supplement, and we have added representative images to Figure 1, per the reviewer’s request. We thank the reviewer for this helpful suggestion. We would be happy to upload raw images to an open-access repository if deemed necessary by the editorial team.

(5) Increasingly, the understanding of heme-dependent roles relies on transient or indirect binding to unsuspected partners, not necessarily relying on a tight affinity and outdating the notion of heme as a static cofactor. Despite impressive recent advancements in the detection of these interactions (for example https://doi.org/10.1021/jacs.2c06104; cited by the authors), a full characterisation of the hemome is still elusive. Sandkuhler et al. deemed it possible but seem to question that heme binding to TANGO2 occurs. However, Sun et al. convincingly showed and characterised TANGO2 binding to heme. It is recommended that the authors comment on this.

We believe it is plausible that TANGO2 binds heme (as do hundreds of other proteins), especially as it has been shown to bind other hydrophobic molecules. However, we also note that a separate paper examining the role of TANGO2 in heme transport posited that GAPDH is the sole heme binding partner for cytoplasmic transport (https://doi.org/10.1038/s41467-025-62819-2), contradicting the originally posited theory of how TANGO2 functions. This is described in the Discussion section and, as noted above, we have added an additional figure to demonstrate our unifying hypothesis for why TANGO2 may be important in the low-heme state, irrespective of any direct effect on heme trafficking.

Additional comments and revisions:

(1) It was suggested that a triple mutant (eat-2; hrg-9; hrg-10) be tested to determine the primary driver of GaPP toxicity. We appreciate this suggestion, but we offer the following rationale for why these experiments were not pursued. The eat-2 mutant, which lacks a nicotinic acetylcholine receptor subunit in pharyngeal muscles, was included solely as a dietary restriction control to illustrate that reduced GaPP toxicity in the hrg-9/10 double mutant could arise from poor feeding rather than defective heme transport. Both eat-2 and hrg-9/10 mutants exhibit markedly reduced feeding but via different mechanisms. In our assays, GaPP survival was inversely correlated with ingestion rate: eat-2 animals, which feed the least, showed the highest survival, while hrg-9/10 mutants showed intermediate feeding and intermediate survival. Consistent with this, eat-2 worms also displayed the lowest ZnMP accumulation.

(2) GaPP solution was added to NGM plates after seeding with OP50. This is now expressly stated in the Methods section (page 15). We would note that Sun et al. mixed GaPP in with NGM in the liquid phase. We would expect that if there were a difference in GaPP exposure due to these different protocols, worms in our experiment would have received higher GaPP concentrations.

“Standard NGM plates were treated with 1, 2, 5, or 10 µM gallium protoporphyrin IX (GaPP; Santa Cruz) after seeding with OP50. Plates were swirled to ensure an even distribution of GaPP and allowed to dry completely.

(3) The manuscript has been reworked to read as more of an independent study rather than a rebuttal of prior work, though the primary objective of validating prior work remains unchanged.

(4) Several technical details of experiments have been moved from the main text to the materials and methods section.

(5) One reviewer noted that the figure numbering should be adjusted. Numbering does not progress sequentially (i.e., 1A…1B…2A…2B) early in the text, because we have opted to consolidate data pertaining to heme analog experiments in Figure 1 and behavioral data in Figure 2.

(6) “Kingdoms” has been changed to “domains” (page 4).

(7) Example images are now included for Figure 1B, as noted above.